# Massive particle interferometry with lattice solitons

Piero Naldesi[1], Juan Polo[2], Peter D. Drummond[3], Vanja Dunjko[4],
Luigi Amico[2], Anna Minguzzi[1] and Maxim Olshanii[4⋆]

**1** Université Grenoble-Alpes, LPMMC, F-38000 Grenoble, France
and CNRS, LPMMC, F-38000 Grenoble, France
**2** Quantum Research Centre, Technology Innovation Institute, Abu Dhabi, UAE
**3** Centre for Quantum Science and Technology Theory,
Swinburne University of Technology, Melbourne 3122, Australia
**4** Department of Physics, University of Massachusetts Boston,
Boston Massachusetts 02125, USA

⋆ maxim.olchanyi@umb.edu

## Abstract

We discuss an interferometric scheme employing interference of bright solitons formed as specific bound states of attracting bosons on a lattice. We revisit the proposal of Castin and Weiss [Phys. Rev. Lett. vol. 102, 010403 (2009)] for using the scattering of a quantum matter-wave soliton on a barrier in order to create a coherent superposition state of the soliton being entirely to the left of the barrier and being entirely to the right of the barrier. In that proposal, it was assumed that the scattering is perfectly elastic, i.e. that the center-of-mass kinetic energy of the soliton is lower than the chemical potential of the soliton. Here we relax this assumption: By employing a combination of Bethe ansatz and DMRG-based analysis of the dynamics of the appropriate many-body system, we find that the interferometric fringes persist even when the center-of-mass kinetic energy of the soliton is above the energy needed for its complete dissociation into constituent atoms.



# 1  Introduction

Bright-soliton interferometry working in the quantum regime has the potential to achieve quantum advantage with an improvement of a device's sensitivity of a factor of a hundred with respect to the standard matter-wave solutions [1]. This idea can be technologically relevant for high-precision force and rotation sensing [1] and measurement of small magnetic-field gradients [2]. At the same time, at more fundamental level, bright solitons separated through beam-splitters are predicted to provide an important route for the creation of macroscopic superposition states [3–6]. Atomtronic devices featuring soliton interferometry were theoretically considered in Refs. [7–9] (see [10, 11] for review and roadmap articles of the atomtronic field). Narrow-barrier beam splitters were studied in Ref. [12–14]. The effect of harmonic confinement on the internal degrees of freedom of a quantum soliton was investigated in [15]. For the splitting process, our major inspiration comes from the work of Castin and Weiss [6, 16]. In their proposal, a bright soliton is scattered off by a Gaussian barrier that is much wider than the soliton width (which was the typical experimental situation at that time). This way, after the scattering, the soliton is in a coherent superposition of being entirely to the left of the barrier and being entirely to the right of the barrier. In such a process, the bright soliton is in a regime where, to an excellent approximation, its behavior can be described as that of an effective point-particle whose mass is that of the whole soliton, and whose position and velocity are those of the center-of-mass (CoM) of the soliton. Following Castin and Weiss, the quantum nature of the problem can be taken into account by assuming that the particle experience an effective barrier potential given by the convolution of the actual barrier potential with the soliton density profile [17, 18] (see also our Eq. 5 below). Such scheme works well because the CoM velocity of the incoming soliton is kept so low that the soliton's CoM kinetic energy is lower than its chemical potential. This ensures that the soliton is energetically protected from breaking into fragments during the collision. Such process, that we will denote 'ionization', is perfectly elastic. In such elastic scattering, the incoming effective particle is quasi-monochromatic: its wavepacket contains a spread of velocities, but the width of the spread is much smaller than the mean incoming velocity of the wavepacket. The quantum transmission probability as a function of the incoming velocity of a strictly monochromatic particle is *almost* a step function (it would be a perfect step-function of the velocity if all the process was entirely classical): as the incoming velocity increases from below the classical

threshold to above it, the transmission probability changes from zero to one, and it does this over a velocity interval (the step width) that is narrow compared to the width of the velocity spread of the wave packet. Under such conditions, the scattering process behaves like classical filtering in Fourier space: to an excellent approximation, the Fourier components that are below the classical threshold are completely reflected and those above it are completely transmitted. Supposing that the velocity spread of the incoming particle includes the classical threshold velocity, the effective particle—and thus the soliton itself—will split coherently into a part that is completely reflected and a part that is completely transmitted. If the mean incoming velocity of the effective particle is exactly at the classical velocity threshold, then the split is 50%–50%.

We should note that the scattering of bright solitons on barriers such that the soliton is typically either wholly transmitted or wholly reflected was experimentally realized and studied in [19]. The authors say that their solitons are too large to form mesoscopic quantum superpositions in the process, but note that such superpositions should be observable for smaller numbers of atoms.

In the present paper, we will work out a specific complete interferometry cycle in which a bright soliton is split on a suitable barrier and then recombined in a harmonic trap. We relax the assumption of perfectly elastic scattering, and the CoM kinetic energy is allowed to be above the ionization threshold. Relying on the recent remarkable progress of light sculpting techniques [20–22], we can make a realistic beam splitter assumption in which the scattering potential can be tightly confined on the spatial length scale of the soliton width [14]. We need to be able to detect the degree of coherence between the reflected and transmitted part of the soliton. For this purpose, we use the coherent splitting described above to construct an interferometer that is sensitive to the presence of an external constant field. The quality of the interferometric fringes is our measure of coherence.

To capture the physics of continuous systems, our numerical analysis is based on DMRG dynamics of lattice bosons in the dilute limit of small filling fractions [23]. Therefore, we are enforced to consider large systems making the DMRG analysis particularly challenging.

Our main result is that, remarkably, the interferometric fringes persist even if the soliton CoM kinetic energy is high enough that it would be energetically allowed for the soliton to break into fragments upon impact with the barrier. Our model is also of experimental relevance as similar schemes have been investigated, albeit they worked in the large particle-number regime and attractive barriers [13]. Finally, we also demonstrate the quantum advantage by considering the quantum Fisher information after splitting, which shows the quantum character of the interferometric scheme.

## 2  Description of the model system

We consider a gas of attracting one-dimensional bosons subjected to various types of external potentials, some of them time-dependent, as required by the interferometric protocol. Our analysis involves both a continuum model employed for the analytical estimates and a lattice model used in the numerical simulations.

We describe $N$ one dimensional bosons of mass $m$ and subjected to attractive contact interactions of strength $g < 0$ by the Hamiltonian

$$\hat{\mathcal{H}}(\omega, \tilde{g}, x_0, F) = \int_{-\infty}^{\infty} dx \left\{ \hat{\Psi}^{\dagger} \left[ -\frac{\hbar^2}{2m} \partial_x^2 + \tilde{g}\,\delta(x) + \frac{m\omega^2(x-x_0)^2}{2} \right] \hat{\Psi} + \frac{g}{2} \hat{\Psi}^{\dagger} \hat{\Psi}^{\dagger} \hat{\Psi} \hat{\Psi} + \hat{\Psi}^{\dagger} F x \hat{\Psi} \right\}, \quad (1)$$

where $\hat{\Psi}$ and $\hat{\Psi}^{\dagger}$ are bosonic annihilation and creation field operators operators satisfying $[\hat{\Psi}(x), \hat{\Psi}^{\dagger}(y)] = \delta(x-y)$. The case $\hat{\mathcal{H}}(0,0,0,0)$ defines the Bose-gas integrable field theory

that is governed by the Lieb-Liniger Hamiltonian [24,25]. The beam-splitting barrier strength is given by $\tilde{g}$; $\omega$ is the frequency of the harmonic potential that is either used to form the interferometer arms ($\omega_{\text{mirror}}$) or for the initial preparation ($\omega_{\text{preparation}}$), and $F$ is a constant force acting on the interferometer, i.e. the "phase object" of the interferometer. The particle number operator is $\hat{N} \equiv \int dx\, \hat{\Psi}^\dagger \hat{\Psi}$. The initial state is chosen as the ground state of $\hat{\mathcal{H}}(\omega_{\text{preparation}}, 0, x_0, 0)$; where $x_0$ is the center of a shifted harmonic potential. Note that the preparation Hamiltonian includes no barrier and no force.

Continuous models can be obtained as lattice systems in the dilute limit [23, 26–28] - see also the appendix. The lattice Hamiltonian leading to Eq. (1) is

$$\hat{\mathcal{H}}_{\text{lattice}}(\kappa, W, L_0, F_{\text{lattice}}) = -J \sum_{j=1}^{L} (\hat{a}_j^\dagger \hat{a}_{j+1} + \text{h.c.}) + \frac{U}{2} \sum_{j=1}^{L} \hat{n}_j(\hat{n}_j - 1) + W \hat{n}_{j_{\text{center}}}$$
$$+ \sum_{j=1}^{L} \kappa(j - (j_{\text{center}} - L_0))^2\, \hat{n}_j + \sum_{j=1}^{L} (j - j_{\text{center}}) F_{\text{lattice}}\, \hat{n}_j, \quad (2)$$

where $F_{\text{lattice}} = Fd$ with $F$ the continuum one and $\kappa$ the spring constant for the harmonic potential on the lattice, with $\kappa_{\text{mirror}}$ being used as the spring constant of the arms of the interferometer and $\kappa_{\text{preparation}}$ for the preparation stage. Here, $\hat{a}_j$ and $\hat{a}_j^\dagger$ are bosonic creation and annihilation operators satisfying $[\hat{a}_j, \hat{a}_k^\dagger] = \delta_{jk}$, and $\hat{n}_j \equiv \hat{a}_j^\dagger \hat{a}_j$. Furthermore, $J$ is the hopping amplitude, $U$ is the onsite interaction constant, and $W$ is the barrier strength. We impose periodic boundary conditions, i.e. we denote $\hat{a}_{L+1} \equiv \hat{a}_1$. We will be working in a sector with a fixed number of particles, i.e. we fix the value of $\langle \hat{N} \rangle$, where the number operator is given by $\hat{N} \equiv \sum_{j=1}^{L} \hat{n}_j$. We assume that number of sites $L$ is odd. The central point of the lattice is given by $j_{\text{center}} \equiv \frac{L+1}{2}$. The length $d$ is the distance between neighboring lattice sites. Similarly to the continuous case, the initial state is the ground state of the following Hamiltonian, $\hat{\mathcal{H}}_{\text{lattice}}(\kappa_{\text{preparation}}, 0, L_0, 0)$: Note that the spring constant is different in the initial state $\kappa_{\text{preparation}}$ and the trapping potential is offset to the left from the center by $L_0 = \text{int}[x_0/d]$ lattice sites. In the following, the analytical calculations will be carried out for the continuous theory Eq. (1). The lattice effects are considered through DMRG of the Hamiltonian Eq. (2).

## 3 Interferometric cycle with a uniform field as a phase object

In figure 1 we show a complete interferometric cycle. The atoms are prepared in the solitonic ground state of a harmonic oscillator of frequency $\omega_{\text{preparation}}$ centered in $-L_0$. At $t = 0$, the preparation confinement is released and the soliton hits the barrier at at $t = T/4$, being $T \equiv 2\pi/\omega_{\text{mirror}}$. Since the barrier is tuned to a 50%–50% splitting, after the collision we end up with an even superposition of half atoms in the left and half atoms in right. At $t = 3T/4$, both wavepackets return to the barrier, where they interfere. We will be interested in the probability of finding the soliton to (say) the left of the barrier. This probability will be sensitive to the presence of a phase object, which will be represented by a uniform force field of intensity $F$. We provide here the main details of the regime of interest for the interferometric protocol. First of all, we work in the regime of weak coupling, ensured by the conditions $|U| \ll 2\pi^2 J$ and $W \ll \pi^2 J$. In such a case there is no need for lattice renormalization, either for the interactions or for the barrier. Furthermore, we choose the parameters in order to ensure that the continuum model applies. This is the case when the healing length

$$\ell = 2\frac{\hbar^2}{m|g|N} = \frac{a}{N}, \quad (3)$$

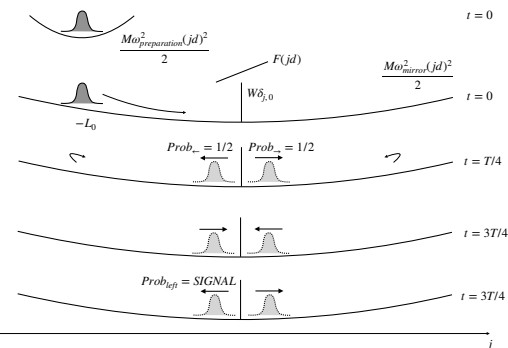

Figure 1: Scheme of the complete interferometric cycle considered in this work. A soliton is prepared at $t = 0$ with an additional harmonic trap centered at $j = -L_0$. The soliton is then released, making it move to the center of the system and collide with the barrier at $t = T/4$. A superposition of the soliton being reflected and transmitted is here created. After the splitting a different phase is imprinted in both branches. The cycle then finishes at $t = 3T/4$ when the two matterwaves interfere.

satisfies $\ell \gg d$. This corresponds to $|U|N \ll 4J$. In the following subsections, we identify all the remaining conditions on the parameters required in each step of the interferometric protocol.

## 3.1 Preparation

The initial soliton will be prepared so that its center of mass (CoM) is in the ground state of a "preparation" harmonic trap, of frequency $\omega_{\text{preparation}}$. Consider the mean kinetic energy of the soliton,

$$E_{\text{kinetic, CoM}} = \frac{1}{2}M\bar{\mathcal{V}}^2, \tag{4}$$

where $M = Nm$ is the soliton mass and $\bar{\mathcal{V}}$ is the mean CoM velocity. Also consider the uncertainty in this kinetic energy $\delta E_{\text{kinetic, CoM}} \approx M\bar{\mathcal{V}}\delta\mathcal{V}$, where $\delta\mathcal{V} = \sqrt{\dfrac{\hbar\omega_{\text{preparation}}}{2M}}$ is the preparation r.m.s. velocity. We will work in the case where the uncertainty in the kinetic energy is finite, but small relative to the mean kinetic energy. It then follows that the preparation r.m.s. velocity is small relative to the mean CoM velocity:

$$\delta E_{\text{kinetic, CoM}} \ll E_{\text{kinetic, CoM}} \Rightarrow \delta\mathcal{V} \ll \bar{\mathcal{V}}.$$

## 3.2 Beam-splitting

The interferometric scheme is based on the splitting of the initial soliton into two copies by means of an atomic beam splitter made by the barrier potential. The condition for a 50%–50% classical filtering as quantum beam-splitting reads [6, 16]

$$E_{\text{kinetic, CoM}} = \max_X V_{\text{soliton-on-barrier}}(X),$$

where $X$ is the CoM position. Note that when the number of atoms $N$ is large, the right-hand side is given by Eq. (B.2), below. The effective potential $V_{\text{soliton-on-barrier}}(X)$ is defined [6] by the convolution of the barrier profile with the soliton density for a center of mass position

localized at $x = 0$, $\rho(x|0)$, according to

$$V_{\text{soliton-on-barrier}}(X) = \int dx\, V_{\text{barrier}}(x - X)\rho(x|0). \tag{5}$$

The latter is known exactly from Bethe Ansatz (we refer the reader to Ref. [18] for a clear derivation). The original result is in Ref. [29].

## 3.3 Mirrors and recombination

We will be using another harmonic trap of frequency $\omega_{\text{mirror}}$ as a "mirror" on each end, to ensure the return of the wavepackets for recombination. The trap frequency and the initial position of the CoM wavepacket, $-L_0$, will conspire to produce the incident energy we need:

$$\frac{M\omega_{\text{mirror}}^2 (L_0)^2}{2} = \max_X V_{\text{soliton-on-barrier}}(X), \tag{6}$$

where, again, if the number of atoms $N$ is large, the right-hand side is given by Eq. (B.2).

## 3.4 The prediction for the fringes, assuming an elastic scattering of the CoM off the barrier.

In the absence of inelastic effects, for a spatially even beamsplitter, the signal will behave as

$$\text{Prob}_{\text{left}}(F) = \sin^2\left(\frac{2NFL_0}{\hbar\omega_{\text{mirror}}}\right), \tag{7}$$

we will derive this in the next subsection. Notice the "quantum advantage factor" $N$ appearing in the argument of the sine function. Due to velocity filtering, there appears a difference between the kinetic energies of the reflected (left) and transmitted (right) wavepackets. However, this does *not* introduce any phase shift on recombination.

In the following, we will derive Eq. (7) in the assumption of no inelastic effects. As in Ref. [6], we assume that sufficiently far from the barrier the CoM wavefunction is accurately described by an effective one-body Schroedinger equation. Let us first define the phase $\phi_{\text{right}}$ as the *total* phase accumulated by the right wavepacket of the CoM wavefunction between the beam-splitting and recombination. It will not include the phase acquired in course of the beamsplitting process itself. The phase $\phi_{\text{right}}$ can be decomposed as

$$\phi_{\text{right}} = \phi_{\text{right}}^{(0)} + \delta\phi_{\text{right}}. \tag{8}$$

Here $\phi_{\text{right}}^{(0)}$ is the phase that would be accumulated if the "phase object" were not present, and $\delta\phi_{\text{right}}$ is the contribution from the "phase object." Analogously, we introduce

$$\phi_{\text{left}} = \phi_{\text{left}}^{(0)} + \delta\phi_{\text{left}}. \tag{9}$$

### 3.4.1 The signal for a given phase difference $\phi_{\text{right}} - \phi_{\text{left}}$

For our interferometer, the signal will be defined as the probability of finding the soliton to the left from the barrier after the recombination. Assuming elastic scattering and no external potential besides the barrier, i.e. neglecting here the effect of the mirror trapping potential, the scattering solution $\psi_{\text{CoM}}(\bar{X})$ for the CoM wave function takes asymptotically the form

$$\psi_{\text{CoM}}(\bar{X}) = \begin{cases} e^{+i\bar{K}\bar{X}} + r e^{-ik\bar{X}}, & \text{for} \quad \bar{X} \to -\infty, \\ t e^{+i\bar{K}\bar{X}}, & \text{for} \quad \bar{X} \to +\infty, \end{cases}$$

where $r$ and $t$ are respectively the reflection and transmission coefficients.

It is easy to show that, after the beam-splitting and recombination, the signal has the form

$$\text{Prob}_{\text{left}} = |r^2 e^{i\phi_{\text{left}}} + t^2 e^{i\phi_{\text{right}}}|^2 .$$

From the conservation of matter, we have $|r|^2 + |t|^2 = 1$, so that, a priori, the family of possible values for $r$ and $t$ is parametrized by three real numbers. But if the scatterer used for both beam-splitting and recombination is spatially even, then $r$ and $t$ are more constrained and their possible values form a family parametrized by two real numbers, $\eta_{\text{e}}$ and $\eta_{\text{o}}$ (see, e.g. [30]). Here 'e' stands for even and 'o' for odd waves. This parametrization is as follows:

$$r = f_{\text{e}} - f_{\text{o}} ,$$
$$t = 1 + f_{\text{e}} + f_{\text{o}} ,$$

where

$$f_{\text{e,o}} = -\frac{1}{1 + i\eta_{\text{e,o}}} ,$$

are the scattering amplitudes for the even and odd waves. The signal now reads

$$\text{Prob}_{\text{left}} = \frac{1}{\left(\eta_{\text{e}}^2 + 1\right)^2 \left(\eta_{\text{o}}^2 + 1\right)^2} \Big\{ -2(\eta_{\text{e}}\eta_{\text{o}} + 1)^2 (\eta_{\text{e}} - \eta_{\text{o}})^2 \cos\left(\phi_{\text{right}} - \phi_{\text{left}}\right)$$
$$+ (\eta_{\text{e}} - \eta_{\text{o}})^4 + (\eta_{\text{e}}\eta_{\text{o}} + 1)^4 \Big\} .$$

We next require that the scatterer be a 50%–50% beam-splitter:

$$|t|^2 = |r|^2 = \frac{1}{2} .$$

There are two disjoint one-parametric families of $r$ and $t$ that have this property, and both can be parametrized by $\eta_{\text{e}}$:

$$\eta_{\text{o}} = -\frac{\eta_{\text{e}} + 1}{\eta_{\text{e}} - 1} , \quad \text{and} \quad \eta_{\text{o}} = +\frac{\eta_{\text{e}} - 1}{\eta_{\text{e}} + 1} .$$

For both families, $\eta_{\text{e}}$ can be any real number except 1 for the first family and $-1$ for the second. While the magnitudes of $r$ and $t$ are now fixed to $1/2$, their phases will depend on the choice of family and the choice of the value of $\eta_{\text{e}}$. Nonetheless, unexpectedly and inexplicably (in the sense that we don't know of any a priori reason why the mathematics had to work out this way), in an interferometer featuring a 50%–50% beam-splitter and a 50%–50% recombiner, the produced signal obeys a universal formula that depends *only* on the phases accumulated between beam-splitting and recombination (and not on the phases of $r$ and $t$, i.e. neither on the choice of the family nor on the choice of $\eta_{\text{e}}$):

$$\text{Prob}_{\text{left}} = \sin^2\left(\frac{\phi_{\text{right}} - \phi_{\text{left}}}{2}\right) . \tag{10}$$

To reiterate, this formula applies to any spatially *even* scatterer, which must be *the same* for both beam-splitting and recombination.

### 3.4.2 The vanishing of the unperturbed phase shift, $\phi_{\text{right}}^{(0)} - \phi_{\text{left}}^{(0)}$

First, let us discuss the effect of the velocity filtering. There will be a difference in kinetic energies between the slow part of the incident wavepacket that gets reflected (left interferometer arm) by the barrier and the fast component that is transmitted (right interferometer arm). A priori, this difference is expected to introduce a phase shift between the arms on recombination. This phase shift will depend on both the width and the shape of the velocity distribution of the incident wavepacket. Moreover, since the "slow" and the "fast" trajectories arriving at the recombiner at the same time would have left the beamspitter at two different instances of time, our interferometer would require a degree of spatio-temporal coherence. Notice, however, that for an interferometer formed by a *harmonic potential*, the latter effect disappears. This is not a coincidence, but an indication that in a harmonic interferometer, velocity filtering does *not* introduce any additional left-right arm phase shift at all. This is indeed the case and can be proven in three ways: (a) quantum-mechanically, (b) semi-classically, using an explicit calculation, and (c) semi-classically, using a variational principle, for small energy differences between the arms only. Moreover, each of the two phases, $\phi_{\text{left}}^{(0)}$ and $\phi_{\text{right}}^{(0)}$, vanish separately.

(a) In a harmonic potential, any initial state $\psi(x, t\!=\!0)$ gets transformed, after a half-period, to a state $\psi(x, t\!=\!\pi/\omega_{\text{mirror}}) = -i\psi(-x, t\!=\!0)$, i.e. to a mirror image of the initial state. No energy-dependent effects are present.

(b) Semi-classically, the phase acquired by the right wavepacket between beam-splitting and recombination will be given by the classical action:

$$
\begin{aligned}
\phi_{\text{right}}^{(0)} &= S_{\text{right}}^{(0)}/\hbar \\
&= \frac{1}{\hbar} \int_{t^{\text{BS}}}^{t^{\text{REC}}=t^{\text{BS}}+T/2} dt \, \mathcal{L}(\bar{X}_{\text{right}}^{(0)}(t), \bar{\mathcal{V}}_{\text{right}}^{(0)}(t)) \\
&= \frac{1}{\hbar} \int_{t^{\text{BS}}}^{t^{\text{REC}}=t^{\text{BS}}+T/2} dt \left\{ \frac{1}{2}M\bar{\mathcal{V}}_0^2 \cos^2(\omega_{\text{mirror}}t) - \frac{1}{2}m\omega_{\text{mirror}}^2(\bar{\mathcal{V}}_0/\omega_{\text{mirror}})^2 \sin^2(\omega_{\text{mirror}}t) \right\} \\
&= \frac{1}{\hbar} \frac{E_{\text{right}}}{\omega_{\text{mirror}}} \int_{t^{\text{BS}}/\omega_{\text{mirror}}}^{t^{\text{BS}}/\omega_{\text{mirror}}+\pi} d\xi \left\{ \cos^2(\xi) - \sin^2(\xi) \right\} \\
&= 0 \,,
\end{aligned}
$$

where $\mathcal{L}(\bar{X}, \bar{\mathcal{V}}) \equiv M\bar{\mathcal{V}}^2/2 - M\omega_{\text{mirror}}^2\bar{X}^2/2$ is the classical Lagrangian for the CoM motion,

$$
\bar{X}_{\text{right}}^{(0)}(t) = (\bar{\mathcal{V}}_0/\omega_{\text{mirror}})\sin(\omega_{\text{mirror}}t)\,,
$$

and

$$
\bar{\mathcal{V}}_{\text{right}}^{(0)}(t) = \bar{\mathcal{V}}_0 \cos(\omega_{\text{mirror}}t)\,,
$$

are the unperturbed CoM trajectory and the corresponding velocity dependence in the right arm, and $M \equiv mN$ is the soliton mass. Further,

$$
\bar{\mathcal{V}}_0 = L_0\omega_{\text{mirror}}\,,
$$

is the velocity of the wavepacket on the beamsplitter, $E_{\text{right}} = M\bar{\mathcal{V}}_0^2/2$ is its energy, and $t^{\text{BS}}$ and $t^{\text{REC}}$ are the beamsplitting and recombination time instances. As one can see, the action *vanishes identically* for any energy of the wavepacket. This derivation can be repeated verbatim for the left interferometer arm. This proof shows that $\phi_{\text{left}}^{(0)} = 0$ and $\phi_{\text{right}}^{(0)} = 0$ separately.

(c) Finally, the absence of the energy dependence of the phase accumulated between the beamsplitting and recombination can be proven variationally, within the semiclassical approximation, for small energy variations. Notice, again, that the slow (left) and the fast (right) wavepackets share the initial and the final points of their trajectories, in *both* space and time. If the energy difference between the trajectories is small, then one of them can be considered a small variation of the other, with fixed space-time end-points. Since the latter trajectory obeys laws of classical mechanics, it must obey principle of least action. Thus, the difference between the two actions (hence between the two quantum phases, in the semi-classical approximation) must vanish to linear order in the amplitude of trajectory variation.

To sum up, we have just showed that

$$\phi_{\text{right}}^{(0)} - \phi_{\text{left}}^{(0)} = 0\,. \tag{11}$$

This property is specific for an interferometric cycle driven by a harmonic potential. Since the right-left energy disparity is conjectured to be unavoidable in interferometry with massive objects [6, 16], a harmonic control of the interferometer arms may provide a remedy for a possible dependence of the fringe position on the energy and shape of the wavepacket.

### 3.4.3 An explicit calculation of the fringe shift due to a uniform field

Finally, we turn to the phase shift accumulated due to the "phase object." Let us compute the phase shift induced by the uniform field $Fx$ on, for example, the right arm of the interferometer. Note that the potential energy correction $Fx$ refers to a single atom. For the CoM, the energy correction *and the resulting quantum phase correction accumulated* must both be multiplied by the number of atoms $N$. The resulting correction to the CoM Lagrangian becomes

$$\delta\mathcal{L}(\bar{X}, \bar{V}) = -NF\bar{X}\,.$$

This amplification, combined with a suppression (which is much harder to achieve) of decoherence of the CoM motion to other degrees of freedom, paves the way to *quantum advantage in particle interferometry*.

Using the principle of the least action, one can easily show (see, e.g. Ref. [31]) that a correction to the arm trajectory introduced by a correction to the Lagrangian does not contribute to a correction to the action, in the first order in $\delta\mathcal{L}$. Thus,

$$
\begin{aligned}
\delta\phi_{\text{right}} &= \delta S_{\text{right}}/\hbar \\
&= \frac{1}{\hbar} \int_{t^{\text{BS}}}^{t^{\text{REC}}=t^{\text{BS}}+T/2} dt\, \delta\mathcal{L}(\bar{X}_{\text{right}}(t), \bar{V}_{\text{right}}(t)) \\
&= \frac{1}{\hbar} \int_{t^{\text{BS}}}^{t^{\text{REC}}=t^{\text{BS}}+T/2} dt\, (-NFL_0)\sin(\omega_{\text{mirror}}t) \\
&= -\frac{NFL_0}{\hbar\omega_{\text{mirror}}} \int_{t^{\text{BS}}/\omega_{\text{mirror}}}^{t^{\text{BS}}/\omega_{\text{mirror}}+\pi} d\xi\, \sin(\xi) \\
&= -\frac{2NFL_0}{\hbar\omega_{\text{mirror}}}\,.
\end{aligned}
$$

An analogous computation for the left arm gives

$$\delta\phi_{\text{left}} = +\frac{2NFL_0}{\hbar\omega_{\text{mirror}}}\,.$$

Putting the two contributions together, we get

$$\delta\phi_{\text{right}} - \delta\phi_{\text{left}} = -\frac{4NFL_0}{\hbar\omega_{\text{mirror}}} . \tag{12}$$

Combining our results in Eqs. (8) to (12), we finally obtain Eq. (7).

## 4   Numerical simulations

Below, we will investigate the effects that soliton 'ionization' has on the interference fringes of a solitonic quantum matter-wave interferometer. In particular, we consider the cases where ionization is energetically allowed (i.e. how large is the CoM kinetic energy of the soliton, Eq. (4), as compared with the interaction effects) affects the degradation of interference fringes from their idealized behavior in Eq. (7).

The full internal energy of the soliton, in the continuous limit (C.L.) and relying on Bethe ansatz perfect string solution, is given by

$$E^{(N)}_{\text{S, interaction}} \overset{\text{C. L.}}{=} -\frac{(N+1)N(N-1)}{24}\frac{mg^2}{\hbar^2} . \tag{13}$$

The gap to the first excitation is

$$E^{(N-1)}_{\text{S, interaction}} - E^{(N)}_{\text{S, interaction}} \overset{\text{C. L.}}{=} \frac{N(N-1)}{8}\frac{mg^2}{\hbar^2} ,$$

the gap to the first two-atom excitation is

$$E^{(N-2)}_{\text{S, interaction}} - E^{(N)}_{\text{S, interaction}} \overset{\text{C. L.}}{=} \frac{(N-1)^2}{4}\frac{mg^2}{\hbar^2} ,$$

the gap to the first three-atom excitation is

$$E^{(N-3)}_{\text{S, interaction}} - E^{(N)}_{\text{S, interaction}} \overset{\text{C. L.}}{=} \frac{3N^2-9N+8}{8}\frac{mg^2}{\hbar^2} ,$$

and the gap to a complete 'ionization' is

$$0 - E^{(N)}_{\text{S, interaction}} \overset{\text{C. L.}}{=} \frac{(N+1)N(N-1)}{24}\frac{mg^2}{\hbar^2} .$$

We will be studying how the stability of the CoM interferometric signal depends on the potential for ionization.[1] Note the following: (a) Refs. [6] and [16] conjecture that being below the single-atom ionization threshold is a prerequisite for a coherence between the transmitted and reflected parts of the CoM wavepacket; (b) Ref. [4] indicates that such coherence is preserved even above the six-atom ionization threshold, in a 100-atom soliton.

Thus, the ionization threshold is

$$E_{\text{kinetic, CoM}} > E^{(N-1)}_{\text{S, interaction}} - E^{(N)}_{\text{S, interaction}} .$$

---

[1]It may seem that breathing excitations constitute another potential inelastic channel. However, curiously, the soliton does not possess true localized excitations: the mean-field breathers are, in reality, unbound. Interestingly, the absence of bound excitations is confirmed in the Bogoliubov approximation. Such a restoration may seem accidental, if Bogoliubov is to be considered as an approximation of mean-filed equations. However, as a linearization of the Heisenberg equations of motion for the quantum field, Bogoliubov can give predictions that are more accurate than the mean-field ones, albeit limited to (perhaps multiple) monomer excitations.

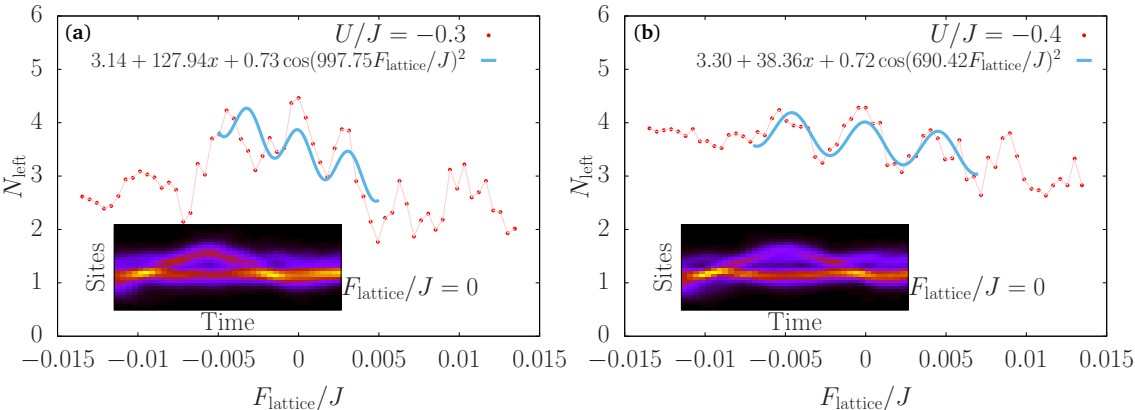

Figure 2: Interference fringes for (a) $U/J = -0.3$ and (b) $U/J = -0.4$ with $N = 6$ and $L = 29$. In (a) the CoM kinetic energy is sufficiently high that total ionization is energetically allowed, while in (b) only two-particle ionization is energetically allowed. Insets show the full time evolution of the interferometric cycle at a force $F_{\text{lattice}}/J = 0$.

This works out to

$$E_{\text{kinetic, CoM}} > \frac{N(N-1)}{8} \frac{mg^2}{\hbar^2}. \tag{14}$$

Meanwhile, the following gives the energy window in which a one-atom ionization is allowed but already a two-atom one is forbidden:

$$E_{\text{S, interaction}}^{(N-2)} - E_{\text{S, interaction}}^{(N)} \geq E_{\text{kinetic, CoM}} > E_{\text{S, interaction}}^{(N-1)} - E_{\text{S, interaction}}^{(N)},$$

which works out to

$$\frac{(N-1)^2}{4} \frac{mg^2}{\hbar^2} \geq E_{\text{kinetic, CoM}} > \frac{N(N-1)}{8} \frac{mg^2}{\hbar^2}. \tag{15}$$

The ratio of the CoM kinetic energy to the ionization thresholds can be tuned by adjusting one or more of $\kappa_{\text{mirror}}$, $U$, and $W$. Once the parameters that determine this ratio are set, we compute $\text{Prob}_{\text{left}}$ (in fact, simply the number of atoms $N_{\text{L}}$ detected to the left of the barrier, since $\text{Prob}_{\text{left}} = N_{\text{L}}/N$) for various values of $F$. This produces a curve like that in Fig. 2, featuring interference fringes. We will present two such curves, corresponding to two different ratios of the CoM kinetic energy to the ionization threshold.

We work in a system of units in which $d = J = \hbar = 1$. In all the runs that we will present, the number of atoms is $N = 6$, the number of lattice sites is $L = 29$, and the initial offset is $L_0 = 7$. In principle, the condition in Eq. (6) links the values of $\kappa_{\text{preparation}}$, $U$, $W$, so that only two of them can be chosen independently. However, after some experimentation, we found that best fringes are obtained if this condition is slightly violated. We ended up varying just the value of $U$ while keeping preparation frequency $\omega$preparation $= 0.0075$ and the barrier strength $W = 0.24$. Furthermore, we also kept $\omega$mirror $= 0.06$.

Under these conditions, we can vary the ratio of the CoM kinetic energy to the ionization thresholds by varying $U$. For each choice of $U$, need to produce a curve of $\text{Prob}_{\text{left}}$ versus $F$. We do this by numerically simulating, using DMRG, the complete interferometric cycle from Fig. 1, at quantum many-body level. We compute the number of atoms $N_{\text{L}}$ detected to the left of the barrier at the end of the cycle.

Having obtained a numerical curve of $N_L$ versus $F$, we compare it to the idealized result in Eq. (7) by fitting the numerical curve to the following function:

$$y(F) = a_1 + a_2 F + a_3 \left[ \cos^2 \left( \omega_{\text{fit}} F \right) \right].$$ (16)

There are 4 fitting parameters accounting for the ($a_1$, $a_2$, $a_3$ and $\omega_{\text{fit}}$,), but we are only interested in $\omega_{\text{fit}}$. According to Eq. (7), $\omega_{\text{fit}}$ should be correspond to $\frac{2NL_0}{\hbar \omega_{\text{mirror}}}$, which works out to 990 for our chosen values of parameters. All the other fitting parameters are merely 'empirical', introduced to account for deviations from the idealized behavior in Eq. (7).

## 5 Results

### 5.1 Interferometric signal

The analytical predictions (7), strictly valid only in the continuum limit, show that for $U/J = -0.3$ and $N = 6$ the soliton should disintegrate onto six individual atoms, leading to a significant suppression of fringes. Increasing the interparticle attractions to $U/J = -0.4$ brings us slightly above the double ionization: the system has enough energy to extract two atoms from the soliton, but not more. In this second case, some part of the system's coherence is preserved and the suppression of fringes is expected to be less strong than for the previous case. Our main result, presented in Figs. 2, shows that the interferometric signal still exists even when a complete disintegration of the soliton onto six individual atoms is energetically allowed.

Insets in Figs. 2 (a) and (b) displays a density plot of the time evolved density distribution of the particles for a chosen set of parameters: $F = 0$, $N = 6$ and $U = -0.3$ and $U = -0.4$ respectively. In the plot is evident how the initial solitonic wavepacket splits and recombines when interacting with the potential barrier. We note that such interferometric sequence does not fully follow the idealized scenario described by the continuum model. A possible cause could come from a combination of effects related to lattice effects introduced in the discretization used in DMRG. One of the most clear differences between the continuum and lattice descriptions is the self-trapping behaviour of the wavepacket close to the barrier potential. This phenomenon, that results in a loss of visibility, has been previously reported [32–35].

Quite remarkably, even above the ionization threshold, in Fig. 2(a) we still resolve the interference fringes of the split solitons. A fitting procedure allows us to extract the oscillation frequency, which is found of the order, but somewhat smaller than the analytical estimate. Note that we only fit around small values of the induced force, as other effects take over for larger values; the main effect being the asymmetric splitting at the first collision due to the force and a delay on the "meeting" point of the split solitons.

Figures 2 and 3 show that the interferometric scheme proposed here is robust against dissociation, and even when the system is energetically allowed to dissociate all particles the fluctuations show us that the state still presents a quantum advantage. Nonetheless, we still need to take into account that one key element is the visibility of the fringes, which are reduced when interactions are smaller (see Fig. 2).

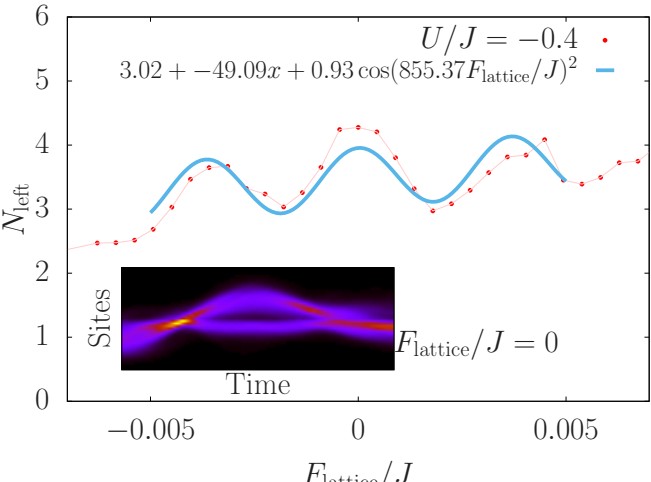

Figure 3: Interferometric fringes for $N = 5$, $L = 41$ and $L_0 = 10$. CoM kinetic energy is sufficiently high that total ionization is energetically allowed. Other parameters are kept the same as in Fig. 2.

## 5.2 Quantum Fisher information

Our interferometric protocol aims to create an optimal spatial macroscopic superposition state $|\phi_{\text{opt}}\rangle$, a superposition of $N$ particles on the left and $N$ particles on right of the barrier:

$$|\phi_{\text{opt}}\rangle \equiv |\psi_{N0,0N}\rangle \tag{17}$$

$$= \frac{1}{\sqrt{2}}(|0\rangle_L |N\rangle_R \pm |N\rangle_L |0\rangle_R). \tag{18}$$

The *quality* of the preparation of this type of state is captured by the the fluctuations of the number of particles over half of the system, defined as:

$$\mathcal{F}(N_L)_{\text{opt}} = \langle \hat{n}_L^2 \rangle - \langle \hat{n}_L \rangle^2, \tag{19}$$

where $\hat{n}_L = \sum_{i=1}^{i_0} \hat{n}_i$. For a pure state such quantity is equivalent to the quantum fisher information and, through the Kramer-Rao bound [36], it gives us the sensitivity of the state to an external applied force. For a NOON state of the form (18), it reads:

$$\mathcal{F}(N_L)_{\text{opt}} = \langle \hat{n}_L^2 \rangle - \langle \hat{n}_L \rangle^2 \tag{20}$$

$$= \frac{1}{2}N^2 - \left(\frac{N}{2}\right)^2 = \frac{N^2}{4}, \tag{21}$$

with the expectation value taken over the state $|\phi_{\text{opt}}\rangle$. For an imperfect cat state these fluctuations are smaller, in the limit of a product state $|\phi_c\rangle = |N/2\rangle_L|N/2\rangle_R$ we would have $\mathcal{F}(N_L)_c = 0$. We will quantify the quality of the interferometric state $|\phi_{\text{int}}\rangle$ we prepare at half of our interferometric protocol, with the parameter $f$:

$$f = \frac{\mathcal{F}(N_L)_{\text{int}}}{N^2}, \tag{22}$$

where now the expectation value is taken over the state after the splitting procedure $|\phi_{\text{int}}\rangle = |\psi(t = t_{\text{int}})\rangle$. If the prepared state has a finite $f$ then the state is of metrological utility and posses a quantum advantage over a classical state.

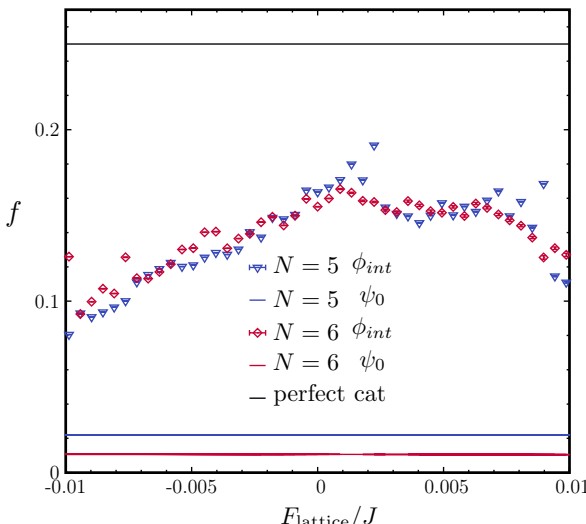

Figure 4: Ratio $f$ as a function of applied force $F_{\text{lattice}}/J$ of both initial state $\psi_0$ and the prepared state $\phi_{int}$. Parameters are set to $N = 5$ (blue) and $N = 6$ (red), $L = 41$ and $L_0 = 10$. Other parameters are kept the same as in Fig. 2.

We now study in Fig. 4, the parameter $f$ as a function of the applied force $F_{\text{lattice}}/J$ for the state generated at one fourth of the full interferometric cycle, $t = t_{int} = T/2$. Despite explicitly calculating this quantity in the regime where particles can be dissociated we are still able to see fringes in Fig. 3 The large value for the parameter $f$ obtained on a broad range of values for the applied force is an indication of having reached the quantum advantage regime.

# 6 Conclusions

We simulated the matter-wave interferometer of the kind originally proposed by Weiss and Castin [6, 16]. Unlike in the original proposal, we considered situations where either partial or complete disintegration of the soliton is energetically allowed. Our main result is that, surprisingly, the interferometric signal survives even when the soliton has enough energy to completely disintegrate. Moreover, the analysis of the quantum Fisher information reflects a quantum advantage of our interferometric setup, even while being well above the aforementioned dissociation threshold.

Our analysis opens the way to further investigations: while our expectation was that the suppression of the interferometric fringes will depend in some simple way on the number of atoms that the soliton is energetically allowed to lose, the numerical simulations did not bear this out. The fringe distortion sequence is more complicated and seems to depend on the structure of the final state.

Two experimental implementation notes are in order. (i) In an experiment, the number of atoms in a soliton fluctuates, leading to fringe degradation. For smaller solitons, a post-selection of the solitons by mass can be sought. For a larger number of particles, one may contemplate using the culling methods [37]. (ii) While the interferometer accuracy will increase with the soliton size (provided the soliton can still be cooled to its ground vibrational state of the "preparation" trap), for large enough solitons, their interaction energy may exceed the transverse vibrational quantum in the waveguide, leading to a collapse [38]. In a typical experiment involving bosonic solitons of $^7$Li atoms [39], collapse occurs at the soliton sizes as large as $N \sim 10^4$ [40].

Finally, in view of the applications, it would be useful to maximize the visibility of the interference pattern at varying the parameters and the conditions of the interferometric setup. This could be realized e.g. by employing a machine-learning scheme. In addition, in this analysis we limited our numerical calculations to be in the dilute regime of the lattice, such that we can rely on the exact solutions of the continuum. However, we can also expect similar behavior in the pure lattice regime [9], while some effect might vary due to the small coupling between COM and relative degrees of freedom occurring in the attractive BHM [41].

## Acknowledgments

**Funding information**    This work was supported by the NSF grants PHY-1912542 and PHY-1607221 and by the ANR-21-CE47-0009 Quantum-SOPHA project.

## A    Continuous vs lattice bosons

The effective one-body mass is given by

$$m = \frac{1}{2}\frac{\hbar^2}{Jd^2},$$

with $d$ being the lattice spacing. Next, let us introduce a two-body scattering length $a$, which is the same in both the continuum and the lattice cases:

$$a = -\frac{2dJ_{K=0}^{\text{rel.}}}{U}\left(1 - \frac{1}{\pi^2}\frac{U}{J_{K=0}^{\text{rel.}}}\right).$$

Here, $J_K^{\text{rel.}} = 2J\cos(Kd/2)$ is the hopping amplitude of the lattice on which the relative motion occurs [42]. For an explanation of the lattice renormalization factor $\left(1 - \frac{1}{\pi^2}\frac{U}{J_{K=0}^{\text{rel.}}}\right)$, see Ref. [17].

The corresponding coupling constant is given in terms of the scattering length in the usual way, from which we may deduce its dependence on the parameters of the lattice model:

$$g = -\frac{\hbar^2}{(m/2)a} \approx Ud.$$

Note that it is not an accident that the lattice and continuum models share the same scattering length. In fact, the effective continuum coupling $g$ is introduced in such a way that it reproduces the lattice scattering length exactly.

In a similar manner, we introduce a scattering length for the particle-barrier interaction,

$$\tilde{a} = -\frac{2dJ}{W}\left(1 - \frac{1}{\pi^2}\frac{W}{J}\right),$$

and the corresponding particle-barrier coupling constant:

$$\tilde{g} = -\frac{\hbar^2}{m\tilde{a}} \approx Wd.$$

The frequency of the "mirror" trapping potential satisfies $\frac{1}{2}m\omega_{\text{mirror}}^2 d^2 = \kappa_{\text{mirror}}$, so that

$$\omega_{\text{mirror}} = \frac{2}{\hbar}\sqrt{\kappa_{\text{mirror}}dJ}.$$

Analogously, the "preparation" frequency is given by $\omega_{\text{preparation}} = \frac{2}{\hbar}\sqrt{\kappa_{\text{preparation}}dJ}$.

## B  Calculation of $V_{\text{soliton-on-barrier}}(X)$ for $N \gg 1$

If the number of atoms is large, the mean-field approximation simplifies the calculations.

The number-density distribution in a soliton is

$$n(x) = \frac{1}{2}\frac{N}{\ell}\text{sech}^2(x/\ell),$$

where $\ell$ is the healing length, Eq. (3). The potential seen by the CoM of the soliton scattered off a $\delta$-function barrier

$$V_{\text{barrier}}(x) = \tilde{g}\delta(x),$$

can be computed as

$$V_{\text{soliton-on-barrier}} \overset{N \gg 1}{\approx} \int dx'\, n(x')V_{\text{barrier}}(X - x') = \tilde{g}n(X), \tag{B.1}$$

see Refs. [6, 16]. This gives

$$\max_X V_{\text{soliton-on-barrier}}(X) \overset{N \gg 1}{\approx} \frac{1}{4}\frac{m|g|\tilde{g}N^2}{\hbar^2}. \tag{B.2}$$

The ionization threshold (14) and the single-atom ionization window (15) respectively become

$$E_{\text{kinetic, CoM}} > \mu_N, \tag{B.3}$$

and

$$2\mu_N \geq E_{\text{kinetic, CoM}} > \mu_N, \tag{B.4}$$

where

$$\mu_N = \frac{N^2}{8}\frac{mg^2}{\hbar^2}, \tag{B.5}$$

is the chemical potential of an $N$-atom soliton, in the mean-field limit. For the 6-atom solitons used in this work, exact formulas must be used starting from about 3-atom excitations.

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
