# Peer review of "Massive particle interferometry with lattice solitons"

_SciPost Physics, doi:SciPost Phys. 15, 187 (2023)_

## Round 1 · Referee Report · Anonymous (Referee 1) · 2023-2-1

Strengths

  1. The paper deals with a proposal of an experimental implementation of an atomic interferometer based on scattering of a Lieb-Liniger-McGuire bright quantum soliton from a sharp potential barrier in the presence of a loose 1D harmonic confinement that provides the recombnation operation. The first strong point of the paper is its sound numerical approach that employs a lattice model.

  2. Based on their analysis, the authors come to a very important conclusion that a (partial or full) disintegration of a bright quantum soliton does not prevent the interferometric signal (the number of atoms in a certain half of the harmonic trap) from being seen.

Weaknesses

  1. However, there is an important issue that has not been properly addressed in the paper, namely, the fluctuations of the number of atoms in a soliton. If we allow for averaging over N fluctuating according, say, to the Poisson law, how strong is the reduction of the interference contrast? I am afraid that the interference fringes on Fig. 2 and 3 will drop well below +/- 1 atom, if we assume the values <N>=5 or 6 as the expectation values for the Poisson distribution (or a Gaussian with a comparable r.m.s.).

  2. It is reasonable to try to increase N (which is not possible numerically, but easy in experiment). But then we may encounter another problem: if the typical energy per particle in a Lieb-Liniger-McGuire bound state becomes larger than the radial trapping frequency, the simple 1D physical picture breaks down and we have to take into account radial excitations (i.e., consider a multiband 1D problem). This issue is also not addressed in the paper.

Report

In general, this paper meets the criteria for SciPost Phys. and can be published after the two weak points mentioned above will have been improved.

Requested changes

  1. The authors should provide their results averaged over a certain distribution of the atom number N. If the Poisson distribution will yield too bad results, the authors should mention the ways to reduce the variance of N, e.g., by a postselection or by some other method. Do the authors expect the laser culling of atoms [Dudarev, Raizen, Niu, PRL 98, 063001 (2007)] to work for this purpose?

  2. The authors should discuss the applicability of the reduction from 3D to 1D in their propoised interferometer or, at least, to give an estimation for the upper limit on N that allows for neglecting radial excitations.

  • validity: good
  • significance: high
  • originality: high
  • clarity: high
  • formatting: good
  • grammar: good

Author:  Piero Naldesi  on 2023-06-07  [id 3713]

(in reply to Report 1 on 2023-02-01)

We thank Referee 1 for the careful reading of our manuscript. The comments/criticisms raised have stimulated us to improve our work. We have carefully revised both the main text and the Supplementary Material. In the following, we provide specific answers to all the comments raised in the Referee 1 reports

Answer to comment n.1:
We agree with the Referee that the number of atoms in a soliton may fluctuate during an experiment, which could potentially result in fringe degradation. To address this issue for solitons composed of a small number of particles, post-selection based on mass may be utilized. However, for solitons with a larger number of particles, culling methods [Dudarev, Raizen, Niu, PRL 98, 063001 (2007)] can be considered.
For such methods, one requires a high level of parameter control in the potentials as well as in the noise produced during the adiabatic protocol followed during preparation. Such control has largely improved in recent years in ultracold atom experiments with new experimental tools such as digital micromirrors, time-average adiabatic potentials, and spatial light modulators see e.g. [Amico et. al. AVS Quantum Science 3 (3), 039201; Rev. Mod. Phys. 94, 041001 (2022)].
We added a paragraph in the conclusion discussing these experimental limitations that could limit the feasibility of our protocol.

Answer to comment n.2:
We would like to thank the Referee for raising the question regarding the limitations of implementation and the potential effects that higher dimensionality can have on the interferometric scheme. To address this concern, we have added a paragraph to the conclusions section of our paper.
The accuracy of the interferometer is expected to increase with the size of the soliton, provided that the soliton can still be cooled to its ground vibrational state in the preparation trap. However, for sufficiently large solitons, the interaction energy may exceed the transverse vibrational quantum in the waveguide, leading to a collapse, see e.g. Gammal, PRA 64, 055602 (2001). In typical experiments involving bosonic solitons of $^{7}$Li atoms see e.g. Luo, PRL. 125, 183902 (2020), where the ratio between normal and transverse confinement is of an order of $10^{-1} \sim 10^{-2}$, collapse occurs for soliton sizes as large as $N \sim 10^{4}$. Such effects of the dimensionality have been explored and discussed in experiments involving the collisions of matter-wave solitons [Nature Physics, 10, 918 (2014)] showing that the 1D regime is accessible under specific conditions.

---

## Round 1 · Referee Report · Anonymous (Referee 2) · 2023-3-27

Strengths

1) The interferometric scheme is clearly explained, and the case for such a study well motivated.

2) The results are very well discussed.

Weaknesses

1) The role of transverse directions should be addressed.

2) An estimate of the effects of the approximations done in the paper to achieve high sensitivity should be as well addressed (see the report).

Report

My general comment is that the paper is clearly written, on a subject of clear interest, and in my opinion it meets the criteria for publication. However I feel that it is important that the Authors consider the role of transverse directions. To be more precise, the setup described by the Authors is based on two main assumptions (leaving apart the temperature): - model (2) is obtained for large optical lattices; and - transverse degrees of freedom have been integrated. I clearly agree that these effects are (or can be made) fairly small and can be taken into account. So, if this was a paper on soliton splitting, I would say that a further analysis would be not needed. But the Authors, as I see from the introduction, are motivated by quantum sensing and I agree with such a motivation. So, one wants to have high sensitivities, with "the potential to achieve quantum advantage with an improvement of a device’s sensitivity of a factor of a hundred with respect to the standard matter-wave solutions". But, do the errors due to Bose-Hubbard approximation and -- more importantly -- due to realistic role of transverse degrees of freedom would prevent from reaching such high sensitivities? To be more explicit, if one has an \omega of tens of Hz, and even an \omega_transverse of tens of kHz (therefore a ratio 10^-3), can one expects to be competitive with other schemes where the relative error of the quantities to be measured is smaller than 10^-3? I am not referring to the instability of the soliton solutions due to transverse degrees of freedom during the dynamics (that it is anyway an issue), but actually on the limit that the unavoidable presence of the transverse directions would put on sensitivity. In other words, if one compares for the numbers given above the full 3D quantum dynamics with the dynamics from the model (2) and finds a relative error 10^-3, that would be excellent - but what about the sensitivity? I do not ask to do a full, systematic study of the impact of approximations on the sensitivity, but I think is important that the issue is mentioned and at least qualitatively discussed.

Requested changes

1) Discuss the effect on the interferometric scheme of the approximations used to write models (2) and (1) (especially the role of transverse degrees of freedom).

  • validity: high
  • significance: good
  • originality: good
  • clarity: good
  • formatting: excellent
  • grammar: excellent

Author:  Piero Naldesi  on 2023-06-07  [id 3714]

(in reply to Report 2 on 2023-03-27)

We express our appreciation to Referee 2 for the thorough examination of our manuscript. The comments and criticisms provided have motivated us to extend and improve our work. We have reviewed both the main text and the Supplementary Material, and below we present detailed responses addressing the comments raised in Referee 2's reports.

Answer to comment n.1:
We thank to the Referee for addressing the query concerning the limitations of implementation and the potential impact of higher dimensionality on the interferometric scheme. To tackle this concern, we have now included a qualitative discussion on the specific impact that the 1D to 2D/3D regimes can have on the interferometric fringes.

The precision of the interferometer is anticipated to enhance as the soliton size increases, provided that the soliton can still be cooled down to its ground vibrational state in the preparation trap. However, when dealing with sufficiently large solitons, there is a possibility for the interaction energy to surpass the transverse vibrational quantum in the waveguide, resulting in collapse (Gammal, PRA 64, 055602, 2001). In typical experiments involving bosonic solitons of 7Li atoms (Luo, PRL 125, 183902, 2020), where the ratio between normal and transverse confinement is approximately 10^(-1) to 10^(-2), collapse occurs for soliton sizes around N ~ 10^4. The dimensional effects have been examined and discussed in experiments exploring the collisions of matter-wave solitons (Nature Physics, 10, 918, 2014), demonstrating that the 1D regime is accessible under specific conditions.

---

## Round 2 · Referee Report · Anonymous (Referee 1) · 2023-8-7

Report

I am satisfied with the authors' answers to my questions. Now it is clear that the paper satisfies the acceptance criteria of SciPost Phys. and can be surely published.

A very small editorial comment. In the newly added text in the Section "Conclusions", the words "For larger particles" should be changed to "For larger numbers of particles" or "For more particles".

---

## Round 2 · Author Response

With this letter, we would like to resubmit our manuscript "Massive particle interferometry with lattice solitons" to SciPost.

We thank the Referees very much for their careful reading of our manuscript and for their positive comments which are answered in the following. We have updated the main text following their suggestions. We are confident that now the manuscript is ready for being published.

Yours Sincerely, Piero Naldesi, Juan Polo, Peter D. Drummond, Vanja Dunjko, Luigi Amico, Anna Minguzzi, and Maxim Olshanii

---

## Round 2 · List of Changes

Besides many small changes to improve the readability of the manuscript, we have discussed two main topics:
1) Number of particles postselection to reduce shot-to-shot errors
2) Influence of the dimensionality on the protocol's sensitivity

---

## Editorial Decision

published